# The Influence of Heat Treatment on Low Cycle Fatigue Properties of Selectively Laser Melted 316L Steel

**DOI:** 10.3390/ma13245737

**Published:** 2020-12-16

**Authors:** Janusz Kluczyński, Lucjan Śnieżek, Krzysztof Grzelak, Janusz Torzewski, Ireneusz Szachogłuchowicz, Artur Oziębło, Krzysztof Perkowski, Marcin Wachowski, Marcin Małek

**Affiliations:** 1Faculty of Mechanical Engineering, Institute of Robots & Machine Design, Military University of Technology, 2 Gen. S. Kaliskiego St., 00-908 Warsaw, Poland; lucjan.sniezek@wat.edu.pl (L.Ś.); krzysztof.grzelak@wat.edu.pl (K.G.); janusz.torzewski@wat.edu.pl (J.T.); ireneusz.szachogluchowicz@wat.edu.pl (I.S.); marcin.wachowski@wat.edu.pl (M.W.); 2Department of Ceramics and Composites, Institute of Ceramics and Building Materials, 9 Postepu St., 02-676 Warsaw, Poland; artur.ozieblo@icimb.pl (A.O.); k.perkowski@icimb.pl (K.P.); 3Faculty of Civil Engineering and Geodesy, Military University of Technology, 2 Gen. S. Kaliskiego St., 00-908 Warsaw, Poland; marcin.malek@wat.edu.pl

**Keywords:** additive manufacturing, selective laser melting, mechanical properties, fatigue properties, heat treatment, hot isostatic pressing, 316L austenitic steel

## Abstract

The paper is a project continuation of the examination of the additive-manufactured 316L steel obtained using different process parameters and subjected to different types of heat treatment. This work contains a significant part of the research results connected with material analysis after low-cycle fatigue testing, including fatigue calculations for plastic metals based on the Morrow equation and fractures analysis. The main aim of this research was to point out the main differences in material fracture directly after the process and analyze how heat treatment affects material behavior during low-cycle fatigue testing. The mentioned tests were run under conditions of constant total strain amplitudes equal to 0.30%, 0.35%, 0.40%, 0.45%, and 0.50%. The conducted research showed different material behaviors after heat treatment (more similar to conventionally made material) and a negative influence of precipitation heat treatment of more porous additive manufactured materials during low-cycle fatigue testing.

## 1. Introduction

Additively manufactured (AM) parts are characterized by very distinctive properties, despite the fact that many different AM technologies where different types of materials are used exist. One of the most important features of parts obtained using AM is a layered structure of the manufactured parts, which significantly affects the anisotropy of mechanical properties during comparison of different building directions [1,2]. This phenomenon is also present in parts processed using laser-powder bed fusion (L-PBF) technologies. Good mechanical properties, a possibility of obtaining geometrically complex parts, and a large spectrum of available alloys allow for the dynamic growth of L-PBF technologies. Use of the AM is seen in aircraft [3,4], automotive [5], armament [6,7], and other solutions that require lightweight structures [8,9,10,11]. The development of new additively manufactured parts that are characterized by their special application needs to be supplied by a proper amount of knowledge about AM material properties and their behavior under different loading conditions. This kind of research project is in the scope of research of many scientific facilities [12,13,14,15,16].

One of the most common materials available for additive manufacturing is 316L steel, which in conventional-manufactured form, is dedicated for applications vulnerable to the adverse effects of chemical and biological factors because of its very good anti-corrosive properties. From a technological point of view, 316L steel belongs to the hard-to-cut materials group—mostly because of its austenitic structure. Additionally, the usage of this steel in medical applications often requires very complex geometry for exact parts. These two factors: applications in a corrosive environment and geometrical complexity, significantly affects material properties that are changing during operation for some specified time.

Despite that there are many available research results connected with mechanical properties of AM parts made of 316L steel, there is still a significant gap in the fatigue properties analysis.

The most significant group of available fatigue test results is connected with high-cycle testing (mostly Wohler’s charts) [17,18,19,20], some works are connected with crack growth analysis of AM parts [21,22,23], but the smallest number of available research results concerns low-cycle fatigue properties, especially with some postprocessing connected with heat treatment or hot isostatic pressing (HIP).

Analysis of the Blinn et al. [24] research work revealed high anisotropy of AM material during fatigue testing, where test parts were manufactured in three different orientations. In samples manufactured vertically to the building plate, it was registered a higher defect tolerance on material damages, which led to higher endurance at lower stress amplitudes.

Mustafa et al. [25] analyzed an AM aluminum alloy, also considering low-fatigue cycle properties, in which the domination of extensive plastic damage beyond grain boundaries was discovered. Similar observations were registered by Romano et al. [26], where authors stated that defect size is the principal cause of variability in the fatigue resistance of the material, even in low cycle fatigue (LCF). Additionally, the authors obtained results where plasticity played an important role in the determination of the fatigue resistance of AM parts.

A different approach was suggested by Bressan et al. [27], in which the authors analyzed an influence of stress-relieving heat treatment on LCF properties of titanium alloys. During their tests, it was observed early sudden material’s weakening, which was correlated with the formation of cracks in internal voids.

Based on two previously-mentioned research works [26,27], a significant influence of porosity in the material volume was stated. It is necessary to understand the reasons for porosity generation in AM parts [28,29]. It is possible to point to two characteristic parts of the AM material structure—porosity in the core of the material and near the outline borders. Regarding voids’ presence in the material volume, an influence of layered structure and connection between fused layers of the material on mechanical properties was also analyzed. Shifeng et al. [30] determined how molten pool boundaries present in the material structure after AM processing affect the mechanical properties of manufactured parts. In the mentioned research, the authors indicated that molten pool boundaries have a significant impact on:‒microstructural slipping during loading;‒macroscopic plastic behavior;‒properties anisotropy;‒low ductility of SLM-processed parts,

On the other hand, Elangeswaran et al. [31] analyzed an influence of surface roughness on the fatigue properties, which indicated a high, negative effect of as-built samples (without surface machining). The main issue was related to unfused powder particles on the samples’ surface, which were a kind of stress raisers.

Microfractographic analysis of the fracture surfaces of the samples allows for the description of the cracking mechanisms. Qualitative fractography plays a special role in explaining the cracking phenomena. It is possible to determine the origin of the crack, the nature of the cracking process. It is often possible to identify defects in material or manufacturing processes.

In this respect, fractal fractography works perfectly well, which deals with the complex aspects of fractures in materials.

Fractal microfractography allows for accurately determining the multidimensional course of the cracking process and is an indispensable element of product quality control [32,33].

During the literature review, it was difficult to find information about the LCF of 316L stainless steel processed using selective laser melting technology with additional analysis, including heat treatment and HIP. Moreover, an LCF analysis mentioned in cited literature included a low portion of microfracture analysis, which is very helpful to understand the damage mechanism of AM metallic parts.

Based on the authors’ own previous research [14,15,34], it was a natural continuation of material analysis connected with the LCF of heat-treated and HIPped parts obtained using the selective laser melting technology (SLM). The main aim of the included research results was to determine the possibility of a void reduction in material volume and describe how it affects LCF properties.

## 2. Materials and Methods

### 2.1. Material

316L steel powder was used for the specimens manufacturing. The material was supplied by Carpenter Additive Company (Carpenter Additive, Widness, UK). It was gas-atomized in the argon atmosphere. Powder particles were characterized by a spherical shape in a diameter of 15–63 μm. Material’s chemical composition is shown in Table 1.

### 2.2. Additive Manufacturing Process Description

The shape of the samples (Figure 1) for fatigue testing was determined based on the recommendations of the ASTM E466 96 standard with considering the dimensions of the building volume of the AM system. For the research, the SLM 125HL system was used (SLM Solutions AG, Lubeck, Germany).

Based on our previous research of the AM process [14,15,34], three-parameter groups were selected, as shown in Table 2. To be consistent with previously published papers—each sample group was named the same as before (Samples: S_01, S_17, S_30).

The S_01 samples were manufactured using producer’s default settings, S_17 samples were manufactured with parameters which gave an increased porosity in [14], S_30 samples were made using almost three times the higher energy density than in default settings and based on Di Wang et al.’s research [35] in which the authors reach good material properties with using mentioned process parameters.

After SLM-processing, the samples were cut out from the build plate, and a sidewall of each sample was milled. The sharp edges of the sidewalls after milling were subjected to a roll-burnishing process to minimize the possibility of stresses arising in that area.

### 2.3. Heat Treatment

Each sample group was subjected to different heat treatment processes to reach different results, which allowed obtaining a higher amount of results and knowledge about the material. Hot isostatic pressing (HIP) was performed on a high-temperature press with furnace Nabertherm VHT 822—GR (Nabertherm, Lilienthal, Germany). Precipitation heat treatment (PHT) was done using the Nabertherm P300 furnace (Nabertherm, Lilienthal, Germany). Each heat treatment process is shown in Figure 2.

The HIP process was performed to reduce the volume porosity and remove the layered structure of the material after the SLM process. Furnace cooling after the HIP process could cause sigma phase generation in austenitic steel [36]—to avoid this and extend the research range, half of the HIPped samples were also subjected to PHT with water cooling, which allowed a significant reduction of the sigma phase formation phenomenon (it generates mostly between 700 and 850 °C).

### 2.4. Low Cycle Fatigue Testing

The scope of fatigue tests of SLM-processed 316L steel included testing of three series of samples produced with the use of three parameters selected during previous tests: S_01, S_17, and S_30. Samples subjected to HIP treatment were named as (S_01H and S_17H), samples subjected to precipitation heat treatment were named as (S_01P, S_17P, and S_30P), samples subjected to both types of heat treatment was named as (S_01HP and S_17HP). Samples geometry was made based on the recommendations of the standards [37,38,39]. The fatigue properties of the tested samples were determined based on the characteristic parameters of the hysteresis loops obtained at:(1)$$\frac{N}{N_{f}}=0.5$$
where:*N*—current number of cycles;*N_f_*—number of cycles to failure.

During testing, the material fracture was assumed as the criterion of sample destruction. The value of the total strain amplitude ε_ac_ was changed sinusoidally with the strain ratio R = 0.1. Fatigue tests were performed with the frequency of load changes f = 0.8 Hz. The values of the total strain were determined based on the previously prepared static tension diagram, shown in [15]. The tests were carried out at five various levels of total strain amplitude: ε_ac_ = 0.30%; 0.35%, 0.40%; 0.45% and 0.50%. For each level, three samples were examined.

Fatigue tests were run on the Instron 8802 servohydraulic testing system and an Instron 2620–603 dynamic extensometer with a gauge length of 25 mm (Instron, Norwood, MA, USA).

### 2.5. Microscopical Analysis

For the fractography analysis, the fracture surfaces from failed specimens were first cut and mounted on an observation stage. The fracture surfaces low-cycle damaged samples were analyzed using the scanning electron microscope (SEM) Jeol JSM-6610 (Jeol, Tokyo, Japan).

## 3. Results and Discussion

### 3.1. Low Cycle Fatigue Properties Analysis

There is significant importance in determining the behavior of material exposed to strain-controlled low-cycle fatigue tests. Mechanisms present in this kind of loading cause local defect formation and consequently lead to the initiation of fatigue cracking. Fatigue strength testing of the SLM-processed materials is particularly justified due to the high possibility of structural imperfections occurrence, most often difficult to avoid without the use of additional postprocessing. A common example of damage in additive manufactured parts is the occurrence of cracks in the zone of increased porosity in the material structure. Such defects locally lower the fatigue strength, especially in thin-walled elements, which are more prone to the formation of porosity due to faster heat dissipation from the melting zone.

Some sample fractures were characterized by non-normative behavior caused by the samples buckling registered during hysteresis loops analysis. The scale of this phenomenon was significantly greater in the case of heat-treated parts. This negative effect was caused by the reduced yield strength of heat-treated samples. Consequently, such cases were not taken into account when compiling the test results.

The variations of stress amplitude σ_a_ with the load loops number are shown in Figure 3, respectively, for each sample group. Additive manufactured samples (S_01, S_17 and S_30) were compared with AM heat-treated parts (S_01H, S_01P, S_01HP, S_17H, S_17P, S_17HP and S_30). All results were compared to the conventionally made material (made of cold-rolled metal sheets—described as P_0 samples).

Concerning the additive manufactured and heat-treated samples, the total strain amplitudes σ_ac_, are characterized by significantly smaller changes in the stress amplitude over the entire range of the number of load cycles N than in the case of non-heat-treated samples. This suggests that the AM material is less prone to cyclic weakening after heat treatment. The mentioned phenomenon is most visible in the case of samples S_01H, S_01P, and S_01HP. In these samples, the stage of cyclic weakening is transient. Additionally, in the range of 7–25% of N_f_ value (ε_ac_ = 0.30, 0.35, 0.40 and 0.45%) and 85–90% (ε_ac_ = 0.50%). The further course of the curves proves the material’s cyclical stabilization. The samples S_01HP and S_17H showed the highest fatigue life.

Research results registered at the lowest value of the total strain amplitude ε_ac_ = 0.30%, shown total fatigue life at a level of 60% in comparison with conventionally manufactured 316L steel. The lowest fatigue life of the tested samples was achieved in samples S_17P (approx. 1550 cycles). It constitutes 16% of the P_0 sample’s fatigue life.

The similar fatigue life of heat-treated samples is noticeable in the samples tested directly after the additive manufacturing process. A significant difference was in the share of the cyclical weakening of the material, which in the case of heat-treated parts is significantly lower than in non-heat-treated samples. Analyzing Figure 3, it can be seen that achieving the same level of total strain amplitude for the conventionally made material (P_0) required a lower stress level than additively produced samples. This effect was due to increased fatigue-durability of conventionally manufactured samples, and at the same time, those samples did not exhibit greater strength at the same level of strain amplitudes as compared to AM samples.

As-built samples were characterized by increased stress levels in mid-life hysteresis loops. This phenomenon was strictly connected with the increased yield strength of as-built samples. In the case of all series of heat-treated samples, very similar courses of the mid-life hysteresis loops were observed during the tests carried out at the total strain amplitude of 0.30%. The coordinates of their peaks indicate the highest stress amplitude values occurrence in the S_30P samples. In the case of the remaining heat-treated samples, it was difficult to state a generalized regularity. The course and position of the peaks of the hysteresis loops (shown in Figure 4) obtained were very similar to the loop of the P_0 reference sample made of conventionally manufactured 316L steel.

To better understand differences between additive manufactured and conventionally made material, an example of the hysteresis loop for the P_0 samples in ε_ac_ = 0.3% strain condition for the peak/valley stressed of characteristic fatigue life cycles was shown in Figure 5. The course of the lower stress values did not change much, while the upper peaks with the number of cycles significantly decrease. The no. 1 cycle in the initial phase showed some irregularity, which may have been related to the compression of the sample in the testing machine holder. This cycle should not be analyzed. The shape of the hysteresis loops was fully regular during the cycles.

Based on Figure 5 observations, it could be stated that the cyclic stress response of investigated specimens can give a qualitative description of cyclic softening phenomena. Similar observations under symmetrical strain control (R = −1) were made by H.S. Ho et al. [40], as well as softening and a fast stabilization presented by S. Romano et al. [26].

The main differences between materials after SLM processing and conventional made are also visible in Figure 3, where additive manufactured material, without any heat treatment weakened in the whole in the entire cycle range. At the same time—conventionally made material (and also additively manufactured after heat treatment) had a visible stabilization at some cycles range. This phenomenon could be connected with the material condition after additive manufacturing—similar to structure and properties after welding (visible melting pools in the structure, increased tensile strength, and decreased elongation at break).

The hysteresis loop surface area is a parameter providing information about material fatigue life as well as the amount of energy necessary for its destruction. Microlevel strains are irreversible plastic deformations, which are related to energy dissipation. It is the main factor causing material damage and the formation of fatigue microcracks. To facilitate measured values of dissipated energy by each sample, hysteresis loop areas were calculated using AutoCAD software (version 2020). The calculated values are shown in the chart (Figure 6). In the case of samples subjected to additional heat treatment, the greatest differentiation of the mid-life hysteresis loops surface areas was observed during the tests at the highest value of the total strain amplitude ε_ac_ = 0.50%.

The biggest hysteresis loop area, higher by 18.5% than the P_0 samples loop area, was found during the S_30P samples tests. Other samples tested at ε_ac_ = 0.50% were characterized by area range from −8.9% to 1.3% compared to the results obtained for the reference P_0 sample. In the case of the remaining samples, the observed differences in the hysteresis loop areas differed by 10–12% from the results obtained for the reference P_0 sample.

Stabilized hysteresis loop analysis allowed to determine the amplitudes of stress and plastic strain. Stress amplitude versus plastic strain amplitude can be described by the following equation:(2)$$log\sigma_{a}=logK^{\prime }+n^{\prime }log\varepsilon_{ap}$$
where:*σ_a_*—fatigue strength coefficient (MPa);*ε_ap_*—fatigue ductility coefficient;*K*′—cyclic strength coefficient;*n*′—cyclic strain hardening exponent.

Calculation results were presented in a chart with a log–log scale in Figure 7.

As it is depicted in Figure 7, almost all AM samples are characterized by higher cyclic stress response at a given strain level, except S_17P samples. The phenomenon of the increased value of this parameter is connected with a higher yield strength of as-built AM parts [15]. Registered higher values of cyclic stress response in AM parts are connected with the LCF testing characteristic, where the strain level was a constant value during the test. Significantly increased yield strength of AM samples gave an increase in cyclic stress response during LCF testing. Heat treatment of AM parts decreased cyclic stress response and made it similar to conventionally made material, and it S_17P this value is even smaller than in conventionally made samples. This phenomenon could be connected with an increased value of porosity in those samples after precipitation heat treatment. Further analysis of S_17P samples’ fatigue behavior was taken into account in microfractures analysis.

Equation (2) for each sample’s group has the following form:$$log\sigma_{a\;S_{01}}=log754.4+0.101log\left(\varepsilon_{ap}\right);$$
$$log\sigma_{a\;S_{01H}}=log459.0+0.06log\left(\varepsilon_{ap}\right);$$
$$log\sigma_{a\;S_{01P}}=log410.8+0.05log\left(\varepsilon_{ap}\right);$$
$$log\sigma_{a\;S_{01HP}}=log602.0+0.11log\left(\varepsilon_{ap}\right);$$
$$log\sigma_{a\;S_{17}}=log631.7+0.09log\left(\varepsilon_{ap}\right);$$
$$log\sigma_{a\;S_{17H}}=log631.7+0.09log\left(\varepsilon_{ap}\right);$$
$$log\sigma_{a\;S_{17HP}}=log427.9+0.07log\left(\varepsilon_{ap}\right);$$
$$log\sigma_{a\;S_{17P}}=log410.8+0.05log\left(\varepsilon_{ap}\right);$$
$$log\sigma_{a\;S_{30}}=log562.1+0.07log\left(\varepsilon_{ap}\right);$$
$$log\sigma_{a\;S_{30P}}=log511.9+0.07log\left(\varepsilon_{ap}\right);$$
$$log\sigma_{a\;P_{0}}=log656.5+0.14log\left(\varepsilon_{ap}\right)$$

Based on the recorded data, further fatigue analysis was followed on the Morrow equation:(3)$$\varepsilon_{ac}=\varepsilon_{ae}+\;\varepsilon_{ap}=\;\frac{\sigma_{f}^{\prime }}{E}{\left(2N_{f}\right)}^{b}+\;{\varepsilon^{\prime }}_{f}{\left(2N_{f}\right)}^{c}$$
where:
$\varepsilon_{ac}$—total strain amplitude;$\varepsilon_{ae}$—elastic strain amplitude;$\varepsilon_{ap}$—plastic strain amplitude;*E*—Young’s modulus;*σ*′*_f_*—fatigue strength coefficient;*ε*′_*f*_—fatigue ductility coefficient;*b*—fatigue strength exponent;*c*—fatigue ductility exponent.

Obtained results (Figure 8) allow stating that heat-treated additive manufactured 316L steel had a significant share of the plastic component during the destruction process; however, in the case of most samples, this share was significantly lower than in the case of non-heat-treated samples. It was particularly visible in the graphs concerning the samples S_17H (Figure 8f), S_17P (Figure 8g) and S_17HP (Figure 8h), where the angles between the lines corresponding to the amplitude of the plastic component ε_ap_ and the amplitude of the elastic component ε_ae_ were much smaller than in the case of the as-built sample S_17 (Figure 8e).

Conventionally produced 316L steel (P_0 samples) was characterized by the lowest share of the plastic component in total deformation of tested samples in comparison to additively manufactured parts (Figure 8k).

### 3.2. Fatigue Fractures Analysis

To better understand material fatigue behavior after a different type of treatment– microstructures from the author’s previous research [15] of samples in each condition (as-built, before and after heat treatment) are shown in Figure 9.

Fractographic observations were made for all parameter groups (S_01, S_17, and S_30) and all heat-treated combinations of samples with additional analysis of conventionally made material. Fracture images of samples tested under two conditions: ε_ac_ = 0.30% and ε_ac_ = 0.50% are shown, respectively. The column “a” shows the entire fracture area and the crack propagation direction, marked using white arrows. In column “b”, there are selected areas of the enlarged fractures.

Based on the fracture’s observations (Figure 10), additively manufactured parts before and after heat treatment were characterized by plastic fractures with fatigue striations, as well as conventionally made P_0 samples. It is worth emphasizing that the morphology of the fracture surfaces of S_01H, S_17H, S_01HP, and S_17HP (all after heat treatment) samples was less complex, which also applied to the reference sample P_0. HIP and precipitation heat-treatment after the previous HIP eliminated the layered structure of additively manufactured parts, which significantly reduced the occurrence of multiplanar cracking. A noticeably greater number of multiplanar cracking was visible in samples subjected to precipitation heat treatment (S_01P, S_17P, S_30P). The significantly different cracking character of those samples may be related to the share of voids in the material structure. Fatigue fractures of the S_17P samples were characterized by a high number of voids and unmelted grains compared to other samples subjected to precipitation heat treatment (S_01P and S_30P). This phenomenon was strictly connected with the high initial porosity of as-built S_17 samples.

The observed structural heterogeneity was caused by the presence of random various voids sizes and share of unmelted grains, which have significantly different properties after heat treatment in comparison to the remaining volume of the material. This phenomenon affects the occurrence of local concentrations of residual stresses in the material structure. Mentioned imperfections significantly affect the development of microcracks inside the material in many parallel planes. Additionally, in S_17P samples, there were registered areas of cracks connection which cause occurring of the local breakage cracks. A significant increase in the share of voids in the S_17P samples completely changed the nature of fatigue cracking of this material. Most of the cracking sources occurred inside the material volume, especially at the most complex geometry of a specific void.

## 4. Conclusions

The performed research of low cycle fatigue of SLM-processed and heat-treated parts allowed drawing the following conclusions:Fatigue life charts created based on the Morrow equation exposed that the strain courses of additively manufactured parts were characterized by a significant share of the plastic component in the process of sample fractures above ε_ac_ = 0.45%;Conventionally made 316L steel (P0) required a lower stress level than additively produced samples, caused by the greater ductility of the conventionally produced material in comparison to AM samples;Microfractures analysis showed that the additively manufactured samples were characterized by a more complex morphology than conventionally produced parts. The observed layered structure of as-built samples affected the occurrence of microcracks and multiplanar cracking;Additively manufactured material, without any heat treatment, weakened throughout the entire load cycle range. At the same time—conventionally made material (and also additively manufactured after heat treatment) were characterized by a visible stabilization at some cycles range;The values of stress amplitude σ_a_ versus the number of load reversals showed much smaller changes of stress amplitude in the entire range of load cycle number N than in the case of non-heat-treated samples (less tendency to weakening of the material). In samples S_01H, S_01P, and S_01HP, the cyclic weakening stage is transient;The obtained strain courses of additively manufactured parts, including samples subjected to additional heat treatment, indicated a significant share of a plastic component in the process of sample fractures.

## Figures and Tables

**Figure 1 materials-13-05737-f001:**
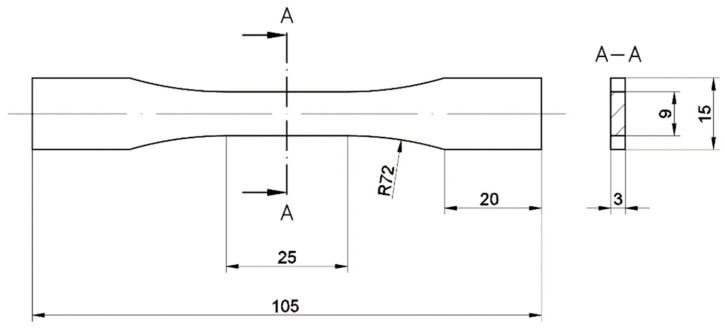
ASTM E466 96 samples shape.

**Figure 2 materials-13-05737-f002:**
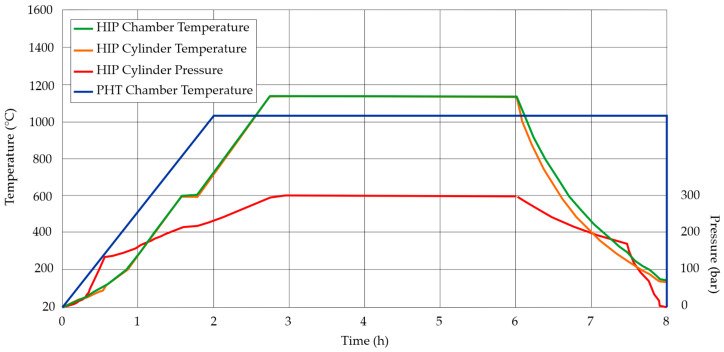
The course of heat treatment processes.

**Figure 3 materials-13-05737-f003:**
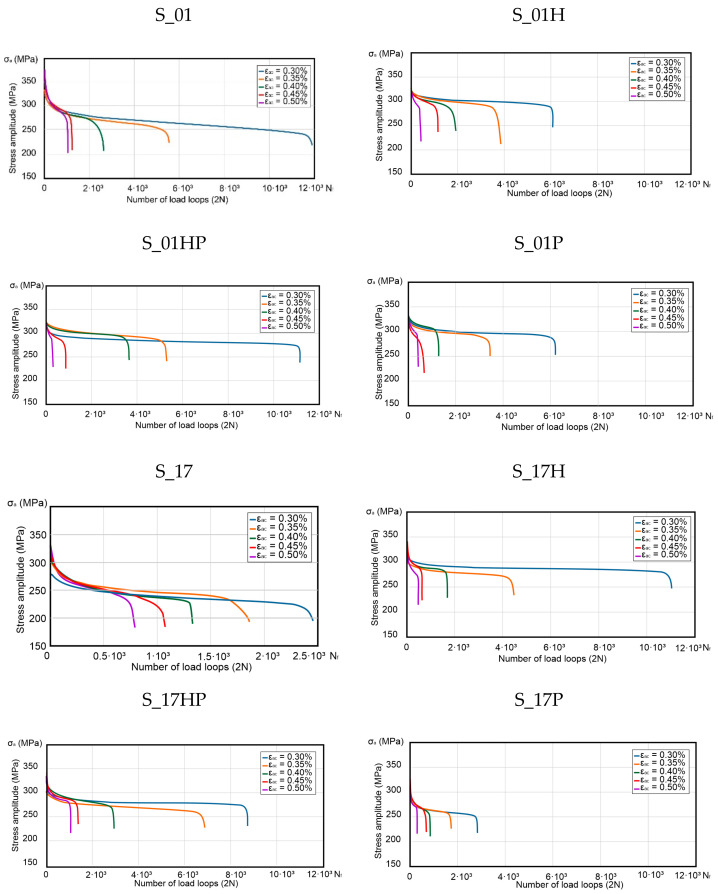
Variation of stress amplitude σ_a_ with the number of cycles for additively manufactured (AM), AM heat-treated parts and conventionally made material.

**Figure 4 materials-13-05737-f004:**
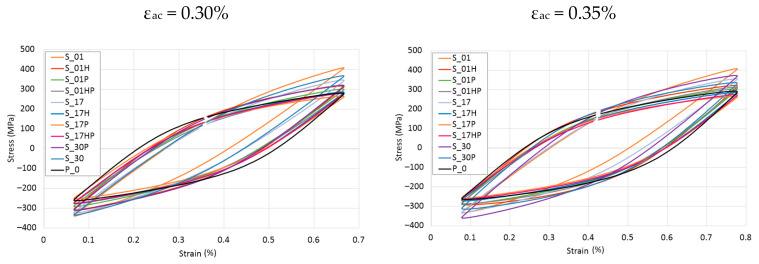
Mid-life hysteresis loops registered for total strain amplitudes ε_ac_ = 0.30, 0.35, 0.40, 0.45, and 0.50%.

**Figure 5 materials-13-05737-f005:**
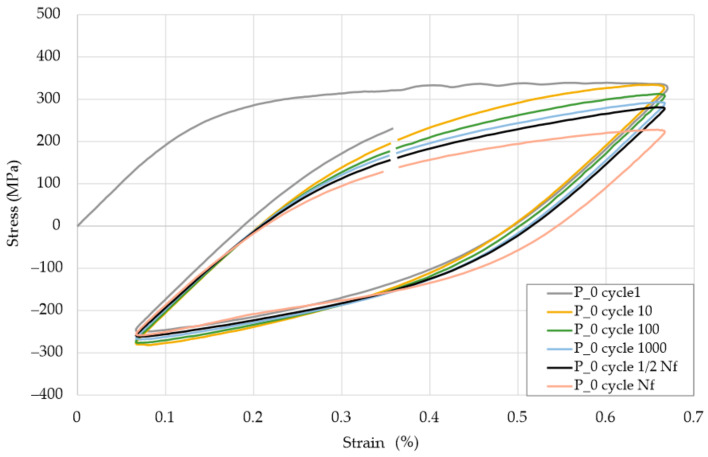
Hysteresis loops for selected cycles for ε_ac_ = 0.3% for characteristic cycles during low cycle fatigue (LCF) testing.

**Figure 6 materials-13-05737-f006:**
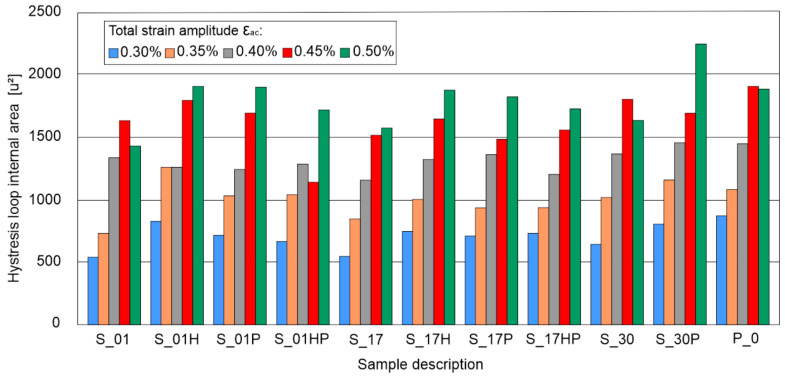
Registered areas of mid-life hysteresis loops (in square units).

**Figure 7 materials-13-05737-f007:**
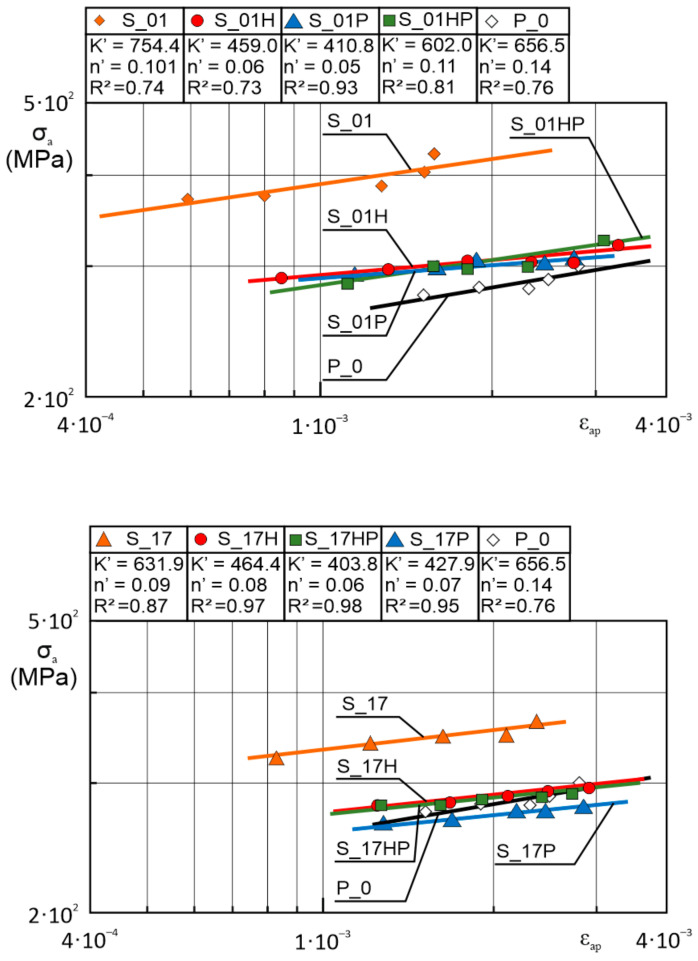
Stress amplitude versus plastic strain amplitude of as-built, heat-treated AM parts compared to conventionally made material stabilized hysteresis loops in log–log coordinates.

**Figure 8 materials-13-05737-f008:**
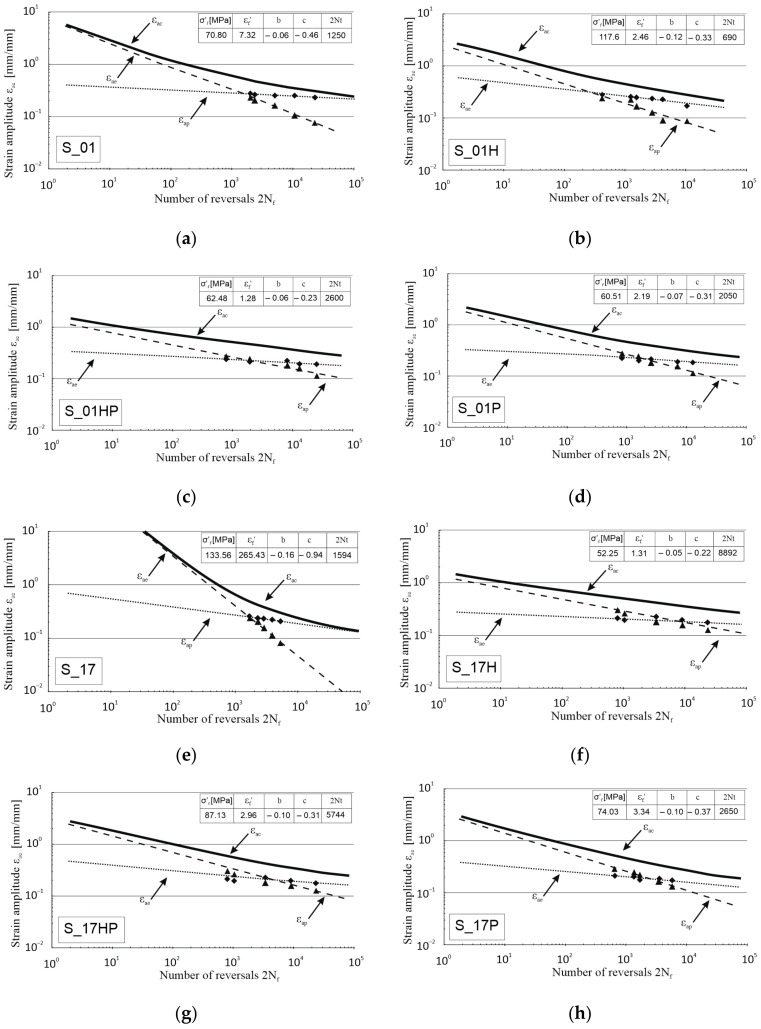
Number of half-cycle reversals vs. strain amplitude in log–log coordinates for as-built samples, heat-treated and conventionally made material ((**a**)—S_01, (**b**)—S_01H, (**c**)—S_01HP, (**d**)—S_01P, (**e**)—S_17, (**f**)—S_17H, (**g**)—S_17HP, (**h**)—S_17P, (**i**)—S_30, (**j**)—S_30P, (**k**)—P_0).

**Figure 9 materials-13-05737-f009:**
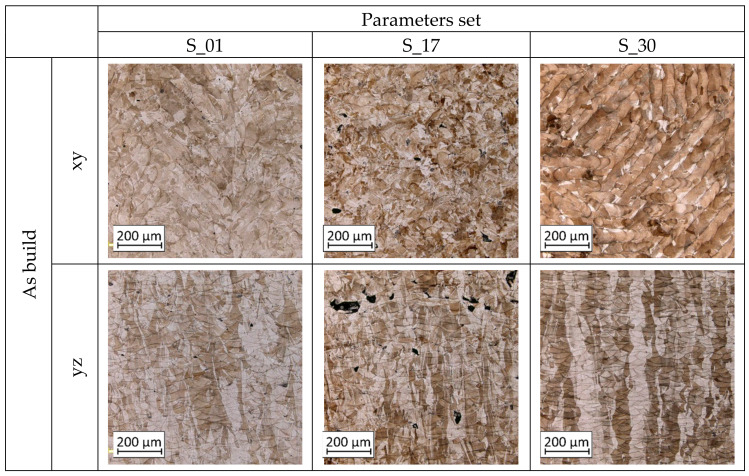
The microstructure of S_01, S_17 and S_30 samples heat-treated using different processes and their combinations [15].

**Figure 10 materials-13-05737-f010:**
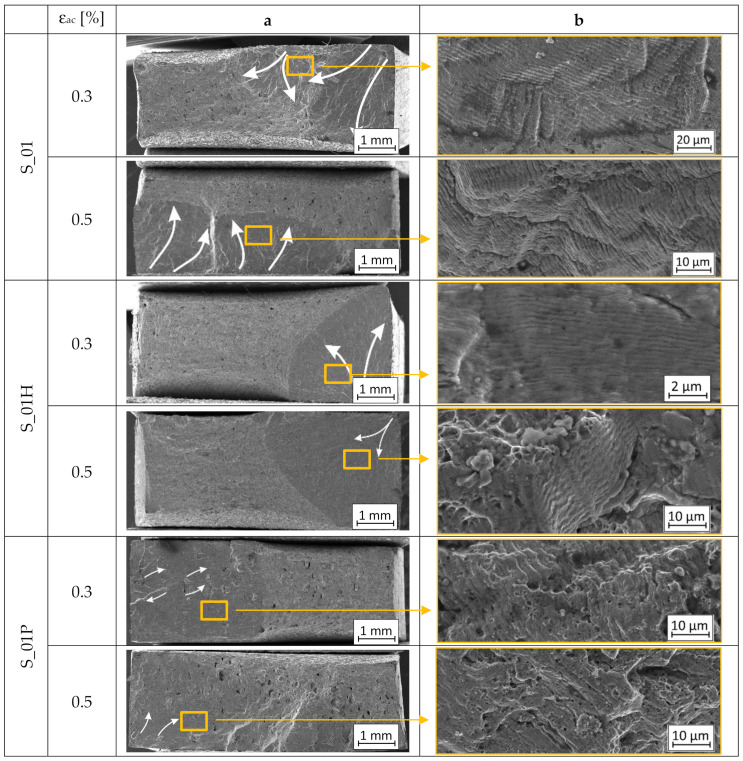
Fractures of low-cycle tested samples ((**a**)—macrostructure image, (**b**)—microfracture with visible fatigue striations).

**Table 1 materials-13-05737-t001:** 316L steel chemical composition.

C	Cu	Mn	Si	O	P	S	N	Cr	Mo	Ni
Weight (%)										
0.027	0.02	0.98	0.72	0.02	0.011	0.004	0.09	17.8	2.31	12.8

**Table 2 materials-13-05737-t002:** Parameter groups used for sample manufacturing.

Parameters Set	Laser Power L_P_ (W)	Exposure Velocity e_v_ (mm/s)	Hatching Distanc eh_d_ (mm)	Layer Thickness (mm)	Energy Density ρ_E_ (J/mm^3^)
S_01	190	900	0.12	0.03	58.64
S_17	180	990	0.13	0.03	46.62
S_30	120	300	0.08	0.03	166.67

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
