# Peer review of "The Influence of Heat Treatment on Low Cycle Fatigue Properties of Selectively Laser Melted 316L Steel"

_materials, 2020, doi:10.3390/ma13245737_

Round 1
Reviewer 1 Report
The manuscript “The influence of heat treatment on low cycle fatigue properties of selectively laser melted 316L steel” proposes propose to find the main differences in 316L steel fracture after AM processing and to analyze how heat treatment affects material behavior during the low cycle fatigue test. .The authors refer that this work is a continuation of a project and previous works, already published.
The work done is clearly important from academical and industrial point of view. The methodology is well designed and the results are interesting. However, the presentation of results has to be improved.
I have some questions / suggestions / changes to propose, not necessarily in this order: -In the theoretical introduction you should mention what are the previous works that you refer to (line 64). -The introduction would be better written and the objectives would be better understood if the authors wrote because they chose 316L steel (even if it has already been explained in previous works). -What the authors call Tables 3,4,5 and 6, are not tables but Figures. Correct them. - In the theoretical introduction, the authors must say something about fracture topography analysis. These inspections can be qualitative (which was what the authors performed) and can be quantitative, as well. Quantitative analysis of the fracture, sometimes called “quantitative fractography”, adds value to the overall study. In this sense, the fractal dimension, for example, is a robust analysis that has the capacity to contemplate the very complex topographic aspects of material fractures, in general, and decode them*. I recommend that the authors add some lines on quantitative techniques, such as fractal dimension, in the topic of the introduction, which is well explained in the cited article.* Horovistiz, A., Campos, K. A., Shibata, S., Prado, C.C.S., Hein, L.R.O, Fractal characterization of brittle fracture in ceramics under mode I stress loading. Mater. Sci. Eng. A. 2010. 527. 4847-4850.
Author Response
Dear Reviewer,
In the beginning, I would like to thank you for taking your time and giving your valuable comments. Regarding your revision, we made proper corrections which were yellow-highlighted in our manuscript. Based on your comments we made corrections as follows:
- “-In the theoretical introduction you should mention what are the previous works that you refer to (line 64).”
- We put proper citations it has been yellow-highlighted and has the following form: “Basing on two previous-mentioned research works [26,27]”
- “-The introduction would be better written and the objectives would be better understood if the authors wrote because they chose 316L steel (even if it has already been explained in previous works).”
- This is a very valuable comment. When we focused on fatigue testing we were less focused on material and just moved directly to a description of fatigue properties in the introduction. We made a proper description to justify material selection in our review (line 44):
“One of the most common materials available for additive manufacturing is 316L steel, which in conventional-manufactured form is dedicated for applications vulnerable to the adverse effects of chemical and biological factors because of very good anti-corrosive properties of this exact steel. From the technological point of view 316L steel belongs to the hard-to-cut materials group – mostly because of its austenitic structure. Additionally, usage of this steel in medical applications require using very complex geometry. Those two factors: applications in a corrosive environment and geometrical complexity significantly affect material properties which are changing during operation for some specified time.
Despite there are a lot of available research results connected with mechanical properties of AM parts made od 316L steel, there is still a significant gap in the fatigue properties analysis.”
- “-What the authors call Tables 3,4,5 and 6, are not tables but Figures.”
We changed all tables containing images and chars to figures.
- “- In the theoretical introduction, the authors must say something about fracture topography analysis. These inspections can be qualitative (which was what the authors performed) and can be quantitative, as well. Quantitative analysis of the fracture, sometimes called “quantitative fractography”, adds value to the overall study. In this sense, the fractal dimension, for example, is a robust analysis that has the capacity to contemplate the very complex topographic aspects of material fractures, in general, and decode them*. I recommend that the authors add some lines on quantitative techniques, such as fractal dimension, in the topic of the introduction, which is well explained in the cited article.”
- We put a proper description of this topic covered by some citations:
“The microfractographic analysis of the fracture surfaces of the samples allows describing the cracking mechanisms. Qualitative fractography plays a special role in explaining the cracking phenomena. It is possible to determine the origin of the crack, the nature of the cracking process. It is often possible to identify defects in material or manufacturing processes.
In this respect, fractal fractography works perfectly well, which deals with the complex aspects of fractures in materials. Fractal microfractography allows for accurately determining the multidimensional course of the cracking process and is an indispensable element of product quality control [32,33].”
- Horovistiz, A., Campos, K. A., Shibata, S., Prado, C.C.S., Hein, L.R.O, Fractal characterization of brittle fracture in ceramics under mode I stress loading. Mater. Sci. Eng. A. 2010. 527. 4847-4850.
- Cabanettes F., Joubert A., Chardon G., Dumas V., Rech J., Grosjean C., Dimkovski Z., Topography of as-built surfaces generated in metal additive manufacturing: A multi-scale analysis from form to roughness, Precision Engineering A. 2018, 52, 249-265
In our research, we used fractographic analysis for cracking behavior analysis without using quantitive methods. The main reason for not using a quantitive method in fractures analysis is a low-cycle character of our fatigue analysis and the presence of a lot of factors (high surface roughness value, porosity, etc.) which significantly affect cracking initiation and could interrupt proper analysis in that kind of analysis.
- To summarize, we hope you found our improvements sufficient. Once again thank you for your comments which could be made a significant improvement to our manuscript.
Yours sincerely,
Authors

Reviewer 2 Report
This is an interesting manuscript presenting experimental low-cycle fatigue results of additively manufactured samples manufactured with the selective laser melting technology and passed through the heat treatment. It shows how heat treatment affects material behavior during low-cycle fatigue testing. These results could be useful for material science researches in case of providing additional explanations and introducing necessary corrections listed in six comments.
Comments:
- Page 2, lines 65, 66: Please clarify the meaning of the sentence: “It is necessary to understand the reasons for porosity generation in AM parts [28,29] in each, characteristic part of the layered structure of the material (porosity in the core of the material, near the outline borders, etc.).” If a comma is placed erroneously, then it should be read something like this:“…in each characteristic part of the layered structure…”.
- Page 3, lines 104-105: Please correct the following sentence: “Basing on own, previous research [14,15,32] for the AM process has been selected three-parameter groups, shown in table 2.” With corrected grammar, it could be rewritten as follows: “Based on our previous research of the AM process [14, 15, 32], three-parameter groups have been selected, as shown in Table 2.”
- Page 4, line 122: Please change the verb tense from past to present in the sentence: “Each heat treatment process was shown in Figure 2.” – “process is shown in Figure 2”.
- The title of Table 3 should be corrected. Change “Variation of stress amplitude (a)” to “Variation of stress amplitude sigma_a”. It is wrongly written in the row before the last of Table 3 results of “S_30P and S_30P”. I guess you presented results for “S_30 and S_30P” – please clarify this question and correct the figure caption “S_30P and S_30P” in Table 3.
- In the text described results of Table 3, please indicate clearly the difference in AM heat-treated parts abbreviated as “H” and “HP”. We believe “H” is heat-treated, but the meaning of “HP” we only could guess, assuming e.g. precipitation heat treatment. Please explain the attribute “P” used in the AM sample names. This attribute is inconvenient because “P_0” in your nomenclature corresponds to “the conventionally made material (made of cold-rolled metal sheets - described as P_0 samples).” It seems that the meaning of attribute “P” in sample names is different. Please explain this point.
- Could you explain why in Table 3 the effect of “HP” is different for AM materials S_01 and S_17? What is the physical phenomenon behind the shortened total fatigue life for AM samples after the heat treatment, and why it is different for S_01 and S_17?

Author Response
Dear Reviewer,
In the beginning, I would like to thank you for taking your time and giving your valuable comments. Regarding your revision, we made proper corrections which were grey-highlighted in our manuscript. Based on your comments we made corrections as follows:
- “Page 2, lines 65, 66: Please clarify the meaning of the sentence: “It is necessary to understand the reasons for porosity generation in AM parts [28,29] in each, characteristic part of the layered structure of the material (porosity in the core of the material, near the outline borders, etc.).” If a comma is placed erroneously, then it should be read something like this:“…in each characteristic part of the layered structure…”.
The sentence has been clarified, now it has a form as follows:
“It is necessary to understand the reasons for porosity generation in AM parts [28,29]. It is possible to point two characteristic parts of the AM material structure - porosity in the core of the material, and near the outline borders.
- “Page 3, lines 104-105: Please correct the following sentence: “Basing on own, previous research [14,15,32] for the AM process has been selected three-parameter groups, shown in table 2.” With corrected grammar, it could be rewritten as follows: “Based on our previous research of the AM process [14, 15, 32], three-parameter groups have been selected, as shown in Table 2.”
We made changed this part based on your advice. Thank you.
- “Page 4, line 122: Please change the verb tense from past to present in the sentence: “Each heat treatment process was shown in Figure 2.” – “process is shown in Figure 2”.
We made changed this part based on your advice. Thank you.
- The title of Table 3 should be corrected. Change “Variation of stress amplitude (a)” to “Variation of stress amplitude sigma_a”. It is wrongly written in the row before the last of Table 3 results of “S_30P and S_30P”. I guess you presented results for “S_30 and S_30P” – please clarify this question and correct the figure caption “S_30P and S_30P” in Table 3.
We made changed this part based on your advice. You are right, it has to be S_30 and S_30P. It is correct now. Thank you.
- In the text described results of Table 3, please indicate clearly the difference in AM heat-treated parts abbreviated as “H” and “HP”. We believe “H” is heat-treated, but the meaning of “HP” we only could guess, assuming e.g. precipitation heat treatment. Please explain the attribute “P” used in the AM sample names. This attribute is inconvenient because “P_0” in your nomenclature corresponds to “the conventionally made material (made of cold-rolled metal sheets - described as P_0 samples).” It seems that the meaning of attribute “P” in sample names is different. Please explain this point.
Please have a look at chapter 2.4. Low cycle fatigue testing where we explained used nomenclature for each group of samples. Our manuscript is a continuation of a project connected with 316L steel behavior testing under different processing and loading conditions. In that case, we want to keep the names of the samples in all our articles.
- Could you explain why in Table 3 the effect of “HP” is different for AM materials S_01 and S_17? What is the physical phenomenon behind the shortened total fatigue life for AM samples after the heat treatment, and why it is different for S_01 and S_17?
The main reason for those differences is connected with parts' initial porosity directly after the additive manufacturing process. It could be also connected with different residual stresses level (which has been taken into account in our previous research). We described a phenomenon of porosity influence in the last chapter of this manuscript, where it has a significant influence on fatigue cracking.
S_01 and S_17 are different from the process parameters point of view. Please have a look at table 2 where we described process parameters.
Regarding your question about the shortened total fatigue life of AM samples after heat treatment, please note that this phenomenon is non-repeatable in different parameter sets. Considering S_01 samples, where increase fatigue life before heat treatment phenomenon is visible, some deeper analyzes are necessary (it is a good idea for additional research). Please have a look at our discussion below in figure 3. AM materials without heat treatment are characterized by a significant share of cyclical weakening which negatively affects fatigue properties, especially in LCF.
Indeed, this is an interesting thing and there is still a huge gap in that kind of research. We have already published seven manuscripts connected with different tests of additive manufactured 316L steel and we are still working on some advanced fatigue analyzes for future publications.
- Summarizing, we hope you found our improvements sufficient. Once again thank you for your comments which could be made a significant improvement to our manuscript.
Yours sincerely,
Authors

Reviewer 3 Report
The authors reported their fatigue test results on AM 316 steel of as-built, Hipped, and Hipped plus precipitation treatment, the information can be value of reference for readers who are interested in AM processing or the material. The reviewer thinks that several aspects should be clarified before publication.
- The authors should show the microstructures of their S01, S7 and S30 AM-build and give the porosity level of these materials. This could be from their previous studies, but would be convenient of the reader of this article to appreciate the microstructural effect.
-
Since the tests were conducted in asymmetrical strain control (with a positive strain ratio), the authors should show the peak/valley stresses, 'cause the stress amplitude alone does not characterize the cyclic process completely, as it is not symmetrical. Under asymmetrical strain-loading, the maximum stress would gradually drop after the first cycle. This is not necessarily due to material weakening but plasticity stabilization. Under symmetrical strain control (i.e., R = -1), Type 316 often cyclic hardens. The authors should compare the behavior with that under symmetrical loading in discussion, at least for P0 condition (look at more reference papers, there are a lot in the literature)
-
As shown in Figure 4, the cyclic stress-strain curves of the AM materials are cyclically stronger than the conventional P0. Add discussion about this phenomena. Actually, here it shows that the untreated AM materials hardened the most. This observation is in contrary of the authors assertion of material "weakening" trend.
-
Revise the conclusion based on the new discussion.
-
Additional editorial comments are annotated by the reviewer, but the authors are encouraged to check the Engliush more throroughly.

Author Response
Dear Reviewer,
In the beginning, I would like to thank you for taking your time and giving your valuable comments. Regarding your revision, we made proper corrections which were green-highlighted in our manuscript. Based on your comments we made corrections as follows:
- “The authors should show the microstructures of their S01, S7 and S30 AM-build and give the porosity level of these materials. This could be from their previous studies, but would be convenient for the reader of this article to appreciate the microstructural effect.”
- We put microstructures from our previous research with self-citation to avoid self-plagiarism. It has been put at the beginning of chapter 3.2. Fatigue fractures analysis
- “Since the tests were conducted in asymmetrical strain control (with a positive strain ratio), the authors should show the peak/valley stresses, 'cause the stress amplitude alone does not characterize the cyclic process completely, as it is not symmetrical. Under asymmetrical strain-loading, the maximum stress would gradually drop after the first cycle. This is not necessarily due to material weakening but plasticity stabilization. Under symmetrical strain control (i.e., R = -1), Type 316 often cyclic hardens. The authors should compare the behavior with that under symmetrical loading in discussion, at least for P0 condition (look at more reference papers, there are a lot in the literature)”
- Regarding your comment, we put an additional chart for P0 samples and made additional descriptions, covered by citations – please have a look at figure 5 and green-lighted text before and after the mentioned figure.
- “As shown in Figure 4, the cyclic stress-strain curves of the AM materials are cyclically stronger than the conventional P0. Add discussion about this phenomena. Actually, here it shows that the untreated AM materials hardened the most. This observation is in contrary of the authors assertion of material "weakening" trend. “
- Figure 4 is now figure 7 (after other reviewers advise). Below that figure we put the following statements (we have also green-highlighted them):
Analyzing Figure 4, it can be seen that achieving the same level of total strain amplitude for the conventionally made material (P0) requires a lower stress level than for additively produced samples. This effect is due to the greater ductility of the conventionally produced material in comparison to AM samples. Besides, conventionally manufactured samples exhibit greater strength as higher cycle numbers are obtained at the same strain levels compared to additively manufactured samples.
- and 5. “Revise the conclusion based on the new discussion. Additional editorial comments are annotated by the reviewer, but the authors are encouraged to check the English more throroughly.”
Conclusions were rephrased and changed parts were green-highlighted. We made a linguistic analysis of the entire manuscript.
6. Summarizing, we hope you found our improvements sufficient. Once again thank you for your comments which could be made a significant improvement to our manuscript.
Yours sincerely,
Authors
Round 2
Reviewer 1 Report
The manuscript has been quite improved. In my opinion it can be published.
Author Response
Dear Reviewer,
Thank you very much for your valuable comment and final acceptance of our manuscript.
Kind Regards,
Authors
Reviewer 3 Report
The authors perhaps missed the point of the reviewer's question # 3 by replaying with the following (also in their revised manuscript, p9)
"Analyzing Figure 4, it can be seen that achieving the same level of total strain amplitude for the conventionally made material (P0) requires a lower stress level than for additively produced samples. This effect is due to the greater ductility of the conventionally produced material in comparison to AM samples. Besides, conventionally manufactured samples exhibit greater strength as higher cycle numbers are obtained at the same strain levels compared to additively manufactured samples."
Strength and ductility are two different aspects of material property. A high ductility material does not necessarily has low strength and vice versa, and the fatigue strains in this study are well below the material ductility. Also, the phrase "conventionally manufactured samples exhibit greater strength as higher cycle numbers are obtained at the same strain levels compared to additively manufactured samples" is not exactly reflect the phenomena. You can say, conventionally manufactured samples are more fatigue-durable than AM samples, but it did not exhibit greater strength (withstand higher stress) at the same strain amplitude levels as compared to AM samples, refer to Fig. 7. Here, strength and durability are two different things, again.
Almost all AM samples except S-17P exhibit higher cyclic stress response at a given strain level, Fig. 7. This is an interesting phenomenon. Therefore, the reviewer would like the authors to give a bit deeper discussion. What's the major contributor to the higher cyclic stress-strain response and what is the major killer of fatigue life (durability)?
Author Response
Dear Reviewer,
Based on your comments we made corrections as follows:
- "Strength and ductility are two different aspects of material property. A high ductility material does not necessarily has low strength and vice versa, and the fatigue strains in this study are well below the material ductility. Also, the phrase "conventionally manufactured samples exhibit greater strength as higher cycle numbers are obtained at the same strain levels compared to additively manufactured samples" is not exactly reflect the phenomena. You can say, conventionally manufactured samples are more fatigue-durable than AM samples, but it did not exhibit greater strength (withstand higher stress) at the same strain amplitude levels as compared to AM samples, refer to Fig. 7. Here, strength and durability are two different things, again.":
We rephrased this sentence - it has been blue-highlighted in our manuscript (lines 215-219), and has the following form now:
"Analyzing Figure 3, it can be seen that achieving the same level of total strain amplitude for the conventionally made material (P0) requires a lower stress level than for additively produced samples. This effect is due to increased fatigue-durability of conventionally manufactured samples, and at the same time, those samples did not exhibit greater strength at the same level of strain amplitudes as compared to AM samples."
2. "Almost all AM samples except S-17P exhibit higher cyclic stress response at a given strain level, Fig. 7. This is an interesting phenomenon. Therefore, the reviewer would like the authors to give a bit deeper discussion. What's the major contributor to the higher cyclic stress-strain response and what is the major killer of fatigue life (durability)?":
We made a proper discussion of achieved results - it has been put directly below Figure 7, and has been blue-highlighted. Its form is as follows:
As it is depicted in figure 7 almost all AM samples are characterized by higher cyclic stress response at a given strain level, except S_17P samples. The phenomenon of the increased value of this parameter is connected with a higher yield strength of as-built AM parts [15]. Registered higher values of cyclic stress response in AM parts are connected with the LCF testing characteristic, where the strain level was a constant value during the test. Significantly increased yield strength of AM samples gave an increase in cyclic stress response during LCF testing. Heat treatment of AM parts decreased cyclic stress response and make it similar to conventionally made material, and it S_17P this value is even smaller than in conventionally made samples. This phenomenon could be connected with an increased value of porosity in those samples after precipitation heat treatment. Further analysis of S_17P samples' fatigue behavior has been taken into account in microfractures analysis.
We hope our corrections met your expectations.
Thank you very much for your comments.
Kind regards,
Authors